ConBGAT: a novel model combining convolutional neural networks, transformer and graph attention network for information extraction from scanned image

Ho Vo Hoang Duy
Vo Quoc Huy
http://orcid.org/0000-0002-9400-7582 Hung Bui Thanh buithanhhung@iuh.edu.vn
Data Science Laboratory/Data Science Department/Faculty of Information Technology, Industrial University of Ho Chi Minh City , Ho Chi Minh , Vietnam
Alatas Bilal
Electronic publication date: 2024 Nov 28
Publication date: 2024
Volume: 10
Electronic Location ID: e2536
Received 2024 Apr 25; Accepted 2024 Oct 30
Copyright: © 2024 Ho Vo Hoang et al.
Copyright year: 2024
Copyright holder: Ho Vo Hoang et al.
License: This is an open access article distributed under the terms of the Creative Commons Attribution License, which permits unrestricted use, distribution, reproduction and adaptation in any medium and for any purpose provided that it is properly attributed. For attribution, the original author(s), title, publication source (PeerJ Computer Science) and either DOI or URL of the article must be cited.
License URL: https://creativecommons.org/licenses/by/4.0/

Keywords: Information extraction, CNN, Bert, GAT, Deep learning, Scanned image

Funding: The authors received no funding for this work.

==============================
Extracting information from scanned images is a critical task with far-reaching practical implications. Traditional methods often fall short by inadequately leveraging both image and text features, leading to less accurate and efficient outcomes. In this study, we introduce ConBGAT, a cutting-edge model that seamlessly integrates convolutional neural networks (CNNs), Transformers, and graph attention networks to address these shortcomings. Our approach constructs detailed graphs from text regions within images, utilizing advanced Optical Character Recognition to accurately detect and interpret characters. By combining superior extracted features of CNNs for image and Distilled Bidirectional Encoder Representations from Transformers (DistilBERT) for text, our model achieves a comprehensive and efficient data representation. Rigorous testing on real-world datasets shows that ConBGAT significantly outperforms existing methods, demonstrating its superior capability across multiple evaluation metrics. This advancement not only enhances accuracy but also sets a new benchmark for information extraction in scanned image.

Introduction

Identifying information within scanned images presents a significant challenge with far-reaching implications for both natural language processing and computer vision. In today’s digital era, the ability to extract data from scanned images is essential across a wide range of applications. Deep learning methodologies have emerged as powerful solutions to this challenge, enabling systems to autonomously recognize and extract relevant information. By utilizing deep learning models (Parveen et al., 2023; Hung & Thu, 2024; Hung, 2023), we can train systems to automatically recognize and extract relevant information from scanned images. These models learn the underlying structures of natural language and image features, enhancing the accuracy and efficiency of information extraction. This automation not only streamlines processes but also reduces the time and effort required compared to manual methods. Furthermore, beyond improving efficiency and accuracy, this capability opens up diverse applications and research opportunities from office automation to the comprehensive management of textual information across image datasets while maintaining standards of transparency and accuracy.

Despite significant advancements in deep learning and computer vision, several challenges still hinder information extraction from scanned images. These challenges arise from the data’s inherent complexity and variability, as well as the limitations of current methods.

One of the most challenging aspects of information extraction is handling unstructured data. Scanned images often contain text in various formats, orientations, and layouts, which complicates the extraction process. Unlike structured data that follows a predictable format, unstructured data lacks consistency, making it difficult to use traditional extraction methods. To extract meaningful insights from such datasets, robust and scalable algorithms are essential. These algorithms must be sophisticated enough to manage the variability in text presentation—such as differences in fonts, sizes, and alignment—while also accommodating potential noise and distortions in the scanned images.

Secondly, there is the challenge of ambiguity and contextual dependencies in information extraction tasks. In complex documents, the meaning of a text segment can be heavily influenced by its context, which might include surrounding text, images, or layout features. For example, a term like “Total” in a receipt scanned image could refer to a numerical value, a label, or a section heading, depending on its position and context within the document. Current models, which often prioritize textual data, may struggle to accurately interpret such terms without considering the broader context of the document. This can result in extraction errors, where incorrect information is identified or important details are overlooked.

Finally, as the volume of digital documents continues to grow, ensuring the scalability and efficiency of information extraction systems becomes increasingly important. Deep learning models, particularly those requiring pre-training on large datasets like LayoutLM (Xu et al., 2020a), are computationally intensive and time-consuming. These models may perform well on specific tasks but often require significant resources to train and fine-tune, which can be a bottleneck when scaling up to handle large datasets or new types of documents. Furthermore, maintaining efficiency while ensuring high accuracy is a delicate balance-improving one often comes at the cost of the other.

Identifying information from scanned images requires combining multiple image processing and natural language processing techniques. In this study, we propose an effective model to identify scanned image information. Our research is inspired by recent research on graph neural networks and information extraction. Our contributions in this research include the following: We propose a new model ConBGAT for the extracting scanned image information. The ConBGAT model has features extracted from combining advanced models CNN in image feature extraction and Transformer-DistilBERT in text feature extraction.

The proposed model utilizes graph modeling techniques to represent the interrelations among regions within text images, enabling the model to effectively capture the complex relationships between components.

Performing comprehensive experiments and multidimensional evaluations on real-world datasets, juxtaposing the performance of our proposed model, ConBGAT, against other existing methods.

Background theory

Convolutional neural networks

Convolutional neural networks (CNNs) are a type of deep learning model particularly well-suited for processing grid-like data structures, such as images. CNNs are widely used in computer vision, natural language processing, and other fields because they effectively capture spatial hierarchies in data through their convolutional layers. The key components of CNNs include convolutional layers, pooling layers, and fully-connected layers with activation functions.

In the context of information extraction (IE), CNNs have demonstrated effectiveness in identifying patterns in text data by treating sequences of words as a 1D grid structure. By applying convolutional filters over word embeddings, CNNs can detect local features that signify specific linguistic elements, such as named entities, relationships, or other relevant text structures. This ability to extract local features makes CNNs highly effective for tasks like sentence classification, named entity recognition, and relation extraction.

The transformer

The Transformer model, introduced by Vaswani (2017), marks a major advancement in deep learning for sequential data tasks like natural language processing (NLP). Unlike recurrent neural networks, which process sequences step-by-step, the Transformer processes entire sequences simultaneously using self-attention. This mechanism allows for efficient parallelization during training and effectively captures long-range dependencies, overcoming key limitations of earlier models.

The Transformer’s key components include self-attention, multi-head attention, positional encoding, feed-forward networks, layer normalization, and residual connections. Its ability to model complex relationships and dependencies in text, combined with efficiency and scalability, makes it highly effective for information extraction from unstructured data. As Transformers evolve, their role in developing more sophisticated and accurate information extraction systems is expected to grow.

Graph neural network

Graph neural networks (GNNs) extend the capabilities of neural networks to handle graph-structured data, where nodes represent entities and edges represent the relationships between them. This makes GNNs particularly suitable for tasks involving relational data, such as social networks, knowledge graphs, and text data where entities and their connections are important.

In a graph, data is represented by nodes (vertices) connected by edges. Each node represents an entity, while edges represent the relationships between these entities. GNNs utilize this structure to learn representations that capture both the features of individual nodes and the connectivity patterns within the graph. Key components of GNNs include message passing, graph convolutional layers, and pooling and readout functions.

In IE, GNNs excel at modeling the non-Euclidean structure of text data, capturing complex entity interactions that are essential for accurate information extraction. They are particularly effective in scenarios where relationships between entities cannot be easily represented by traditional sequential models.

Related works

Graph neural network

GNNs are a specialized class of machine learning models tailored for processing graph-structured data (Zhou et al., 2020; Goyal & Ferrara, 2018; Allamanis, Brockschmidt & Khademi, 2017; Kipf & Welling, 2016; Wu et al., 2020). These algorithms are highly effective at learning representations of nodes, edges, and entire graphs. GNNs provide both scalability and versatility, making them ideal for analyzing complex structures across a variety of domains, including social networks, transportation systems, and other interconnected systems.

GNN methods typically fall into two primary categories based on their architectural principles: spatial and spectral. Spatial methods draw inspiration from the success of CNNs in image processing. They operate by capturing local neighborhood interactions to update node representations (Duvenaud et al., 2015; Li et al., 2016). On the other hand, spectral methods rely on spectral graph theory (Shuman et al., 2013), utilizing graph Laplacians to define convolution operations within the graph domain (Defferrard, Bresson & Vandergheynst, 2016; Sahbi, 2021).

In today’s landscape, research on GNNs continues to evolve, aiming to bolster the model’s performance and broaden its applications across diverse domains. A plethora of GNNs variants have emerged to tackle the complexities of graph data. Among them, graph convolutional networks (GCNs) (Zhang et al., 2019; Chen et al., 2020a; Yang et al., 2020; Pei et al., 2020; Chen et al., 2020b) stand out as a fundamental and widely adopted form. GCNs employ a graph convolution mechanism to propagate information across vertices and edges within the graph. By integrating vertex features and graph structure, GCNs facilitate classification or prediction tasks on graphs.

Another notable variant, Graph Sample and Aggregated (GraphSAGE) (Hamilton, Ying & Leskovec, 2017; Ding et al., 2021; Xiao, Wu & Wang, 2019; Rong et al., 2019; Wang & Vinel, 2021), employs a strategy of sampling and aggregating information from neighboring vertices to update each vertex’s features. This approach enables GraphSAGE to glean insights into the overall graph representation, thus enhancing its capability to handle large-scale graphs effectively.

Furthermore, graph attention networks (GATs) (Veličković et al., 2017; Brody, Alon & Yahav, 2021; He et al., 2023; Busbridge et al., 2019; Sun et al., 2023; Belhadj, Belaïd & Belaïd, 2023; Belhadj, Belaïd & Belaïd, 2024) leverage the attention mechanism to assess the significance of neighboring vertices for each vertex in the graph. Through weighting the information from neighbors, GATs prioritize important vertices, enabling dynamic processing of the graph.

Information extraction

Several machine learning methods address Information Extraction from informal texts. Rote learning memorizes instances of information fields, while term-space learning uses Naive Bayes to estimate the likelihood of a text segment being an instance based on word statistics. Learning abstract structure combines grammatical inference with term-space learning to enhance boundary detection. Relational learning applies rule-based algorithms to classify text segments, using both token features and their relationships (Freitag, 2000; Kaddas et al., 2023).

In recent years, concurrent with the evolution of GNNs models, there have been notable strides in the field of information extraction from scanned images. An increasing number of research endeavors leverage these GNNs models in tandem with diverse methodologies to extract information from images. In our study, we draw upon several pertinent investigations in this domain.

One such study focuses on the extraction of information from invoices using a spectral graph convolutional network (Hung, 2022). This article offered a comprehensive overview of techniques for extracting information from invoices, encompassing template-based and natural language processing approaches. Additionally, the article delineates the advantages of employing spectral graph convolutional networks and elucidates how they can effectively address the challenge of information extraction from invoices.

In other work, Lohani, Belaïd & Belaïd (2019) presented an innovative invoice reading system employing GCNs. This system demonstrates remarkable accuracy in reading invoices, even when confronted with diverse layouts. By harnessing GCNs, the system effectively learns both the structural and semantic information inherent in invoice entities. Notably, the system operates without necessitating any predefined invoice format information.

Zhao et al. (2019) proposed the CUTIE model, a Universal Text Information Extractor utilizing CNNs to comprehend document content. However, this model exhibits several limitations: it relies heavily on structured data, prioritizes textual content over graphical representations, struggles to adapt to new data, and lacks interpretational capabilities.

Yu et al. (2021) introduced the Processing Key Information Extraction (PICK) model, designed specifically for extracting key information from text documents.

While many of the research articles and methodologies discussed utilize GCNs for information extraction and graph representation learning, significant constraints in graph processing still exist. These limitations often arise from dependencies on the underlying graph structure of the input data. When input data lacks a well-defined graph structure, establishing relationships between entities becomes difficult. Additionally, large training datasets can complicate pattern recognition, and biased data may result in the learning of inaccurate models.

Optical character recognition

Optical character recognition (OCR) stands as a pivotal technology applied across various domains, ranging from natural language processing to automation. Its capability to convert text images into machine-readable formats significantly reduces the time and labor involved in manual data entry processes.

Tesseract (Smith, 2007), an open-source OCR engine developed by Google, stands out for its robustness and high accuracy in recognizing a wide range of languages. This versatile tool finds extensive application in various practical scenarios, including street sign recognition, digitizing paper documents, and streamlining business processes.

EasyOCR (Liao et al., 2022), developed by OpenCV, is another open-source OCR engine known for its user-friendly interface and swift performance. Capable of recognizing multiple languages, EasyOCR is frequently employed for personal tasks such as converting physical documents into digital formats and QR code recognition.

In this study, we propose a new ConBGAT model for information extraction from scanned images. We utilize advanced models to extract image and text features using CNN and DistillBERT, and then train these features using the GAT model to address the problem of extracting information from scanned images. The process involves identifying regions containing text and assigning appropriate labels to each. We then perform a detailed evaluation of our model’s performance, comparing it with other models using real-world datasets.

Proposed methods

The proposed ConBGAT model

We begin by identifying text regions within the input scanned image. Subsequently, we extract features by embedding both image and text features obtained from a model that combines CNNs and DistilBERT. These features are then utilized to construct a graph representing the input data. The GNNs model is trained on this graph-modeled data, enabling classification of nodes within the graph. Finally, we leverage the trained model to extract text entities classified and predicted by the GNNs learning models, presenting the resulting text information as the output. Our proposed model is presented in Fig. 1.

Figure 1 The proposed ConBGAT architecture with two main components: Feature extraction and graph neural networks model.

Extraction and graph neural networks model

Figure 1 illustrates the proposed ConBGAT architecture, which includes several key components. The model’s input is a scanned image containing information to be extracted. The Word Separator identifies text regions and creates bounding boxes to be fed into the GAT model. For text detection, the CRAFT model (Baek et al., 2019) is employed to predict character bounding boxes, which identify the text regions. OCR tools like Tesseract and EasyOCR are then used to recognize and extract text content from these regions. Feature extraction involves two processes: DistilBERT generates embeddings for the extracted text, leveraging its smaller size and faster processing speed while maintaining good performance, while ResNet50 (He et al., 2016) extracts image features from the text regions using its deep convolutional neural network capabilities. The extracted text and image features are combined to form nodes in a graph, with edges determined by the relative positions of the text regions in the image. The GAT then classifies the nodes, utilizing an attention mechanism to enhance the representations of each node by considering the importance of neighboring nodes. The final output of this process is the extraction of text entities, such as company name, address, date, and total amount, from the scanned image based on the classification results from the GAT.

Word recognition

This section comprises two primary tasks: Firstly, we identify the bounding boxes for each word region within the input scanned image, a process known as text detection. Subsequently, we extract the content of these words to facilitate feature extraction. The detailed steps are outlined below:

Text detection: We utilize the Character Region Awareness for Text Detection (CRAFT) model (Baek et al., 2019) in this study. CRAFT is specifically designed to identify text containers by leveraging character features, thereby achieving high performance, particularly with texts exhibiting complex shapes. The model employs an attention-based mechanism to predict the container for each character within the text. In this study, we leverage established OCR tools to identify text regions, as outlined in the text detection section. Specifically, we employ two prominent OCR engines: Tesseract (Smith, 2007) and EasyOCR (Liao et al., 2022). These tools are renowned for their robust support in text recognition tasks.

Feature extraction

Feature extraction here is to create features for the nodes (nodes, vertices) of the graph. In this study, the nodes of the graph are bounding boxes, which are areas containing text information identified in the text detection section above. We use two features for the nodes: image features and lexical features. We present details in Fig. 2. First, from the bounding boxes defined above, we use two deep learning models to extract features: RestNet50 (He et al., 2016), a CNNs model for extracting features from images, DistilBERT model (Sanh et al., 2019) for extracting text content features in the bounding box. Character is recognized by text recognition toolkits and integrate to nodes features. The definition of edges of the graph will be presented in detail in the next section.

Figure 2 Feature extraction methods for text and image feature extractions.

Figure 2 illustrates the feature extraction process in the ConBGAT model, highlighting key components such as bounding boxes, text features extraction, image features extraction, and the combination of embeddings. Bounding boxes, which are regions containing text information identified during the Text Detection process, correspond to nodes in the model’s graph. Text features are extracted using DistilBERT, generating text embeddings from the recognized text within each bounding box. Simultaneously, image features are extracted using a CNN (ResNet50), and a fully connected layer is applied to align the output size with the text embeddings. The text and image embeddings are then combined through addition, forming a new representation that serves as the node’s features. These combined features are subsequently used as input for the GNNs model, demonstrating how text and image features are integrated from bounding box regions to create input features for the GNNs in the ConBGAT architecture.

Text features extraction: We utilize word embeddings to transform text into vectors that represent the identified words within the text. These vectors are then combined with image features to create a comprehensive feature matrix for each node in the graph. This process enables the model to understand both the textual context and the relevant numerical features, thereby enhancing its predictive and classification capabilities. In our study, we utilize the pre-trained DistilBERT model to generate vector representations for text sentences. DistilBERT, introduced by Sanh et al. (2019), serves as a more compact alternative to BERT (Devlin et al., 2018), offering comparable performance. DistilBERT is developed through a compression process known as ‘distillation,’ where a new model (referred to as the child model) is trained on predictions made by a larger model (the parent model). This process enables the child model to capture the essential features of the parent model without the need for extensive dimensions or parameters.

DistilBERT offers several advantages over BERT, including its reduced size, faster training and inference times, lower resource requirements, and cost-effectiveness, while maintaining equivalent performance capacity.

For a set of text in a document, we combine them according to coordinates from top to bottom and from left to right to form a character string. Given a string tseqk=(a1(k),a2(k),...,ai(k)), text embeddings of a sequence tseqk is represented by Eq. (1).

(1) TEmb0:i(k)=DistilBERT(a1:i(k);ΘDistilBERT)

where a1:i(k)=[a1(k),a2(k),...,ai(k)] is the input sequence padding with a1(k)=[CLS]. The [CLS] token, used for capturing full sequence context, was first introduced by Devlin et al. (2018). TEmb1:i(k)=[TEmb1(k),TEmb2(k),...,TEmbi(k)]iϵℝi∗dmodel represents the output sequence embeddings, dmodelis the dimension of the model. TEmbi(k)represents the ith result of the pre-train model DistilBERT for ith document. ΘDistilBERTrepresents the parameters of the pre-train model DistilBERT. Each sentence is encoded independently, we get the text embeddings of document β with η sentences or text paragraphs. We define it according to Eq. (2).

(2) TFE=[TEmb1:i(1),TEmb1:i(2),…,TEmb1:iη]

Image feature extraction: We used CNNs (O’Shea & Nash, 2015) for image embeddings. Given a set of image fragments created from these bounding boxes iseqk=(b1(k),b2(k),...,bi(k))for each text area in the pre-determined image, it will then be fed into the CNNs model to perform feature representation and calculation for each box. Image embedding is represented by Eq. (3).

(3) IEmb1:i(k)=CNN(b0:i(k);ΘCNN)

where b1:i(k)=[b0(k),b1(k),...,bi(k)]denotes the input image segments appending with b0(k)=fullimg. We utilize b0(k) to represent the overall structural characteristics of the document image. bi(k)ϵℝH∗W∗3 represents ith image segment of kth document, H and W are high and width of image respectively. IEmb0:i(k)=[IEmb0(k),IEmb1(k),…,IEmbn(k)]ϵℝn∗dmodel represents the output image embeddings, dmodel is the dimension of the model. We use a variant of CNN for this image embeddings viz Resnet50 (He et al., 2016) and a fully connect class that resizes the output according to the size of the dmodel. IEmbi(k) represents the ith output of CNN model for the kth document. ΘCNNis parameters of CNN model. By independent encoding, we can get the image embedding of the document β. We define it according to Eq. (4).

(4) IFE=[IEmb1:i(1),IEmb1:i(2),…,IEmb1:i(k)]

After extracting text features (TFE) and image features (IFE), we combine these embeddings to create a new representation by partially adding these features together by Eq. (5). The features are then used as input nodes for the our GNNs model.

(5) Υ¨=TFE+IFE

Graph modeling

As introduced above, in this graph modeling section, we identify the edges of the graph and calculate the relative distances between boxes in the left, right, top and bottom directions if they exist, if they exist. Values that do not exist will be set to 0. Figure 3 shows an overview of the relative distance between boxes on the image.

Figure 3 Relative distances between boxes in the left, right, top and bottom directions on the image.

The relative distance indicators between boxes can be expressed as Eq. (6).

(6) DL=(RIGHT(Boxleft)−LEFT(Boxroot))WIDTHimageDT=(BOTTOM(Boxtop)−TOP(Boxroot))HEIGHTimageDR=(LEFT(Boxright)−RIGHT(Boxroot))WIDTHimageDB=(TOP(Boxbottom)−BOTTOM(Boxroot))HEIGHTimage

where DL, DT, DR, DB corresponding to the relative distances left, above, right, and below the word Boxroot (root box) to neighboring boxes.

The above parameters will be calculated based on the coordinates of the bounding boxes. These bounding boxes have been previously defined (in the text detection section). For example, with DB will be equal to the distance from the original box to the box below and divided by the height of the image HEIGHTimage. With other parameters, perform similar calculations. Figure 4 shows in detail the results we achieved after constructing graphs for the data.

Figure 4 Scanned image after constructing graphs for the data.

Graph attention network models

In this study, we adopt a variant of GNNs known as the GAT for our analysis. The GAT model is selected for evaluation and comparison across both the SROIE dataset and our collected dataset.

GAT operates as a type of GNNs that leverages the attention mechanism to ascertain the significance of neighboring vertices within the graph. Through weighted aggregation of neighbor information, GAT prioritizes crucial vertices and dynamically processes the graph. Notably, GAT demonstrates superior performance in various tasks, including classification, regression, and link prediction.

While GAT shares the foundational architecture of GCNs, it distinguishes itself by employing attention calculations instead of conventional convolutions to determine vertex importance. Below is the generalized formula of GAT for a propagation in the model.

Suppose the input of the model has N vertices and each vertex u will be represented by a feature vector hu∈ℝF, with F is the dimensionality of the feature.

Compute the attention weight for each pair of vertices u and v.

(7) euv=LeakyRELU(aT[Whu||Whv])

where, a∈ℝ2F, is a learned weight vector W is a learned weight matrix and || is the operator that joins two vectors. Soften the attention weights to sum to 1.

(8) αuv=exp⁡(euv)∑v∈Nu⁡exp⁡(euk)

where Nu represents the set of adjacent vertices of vertex u.

Output of each u vertex.

(9) hu′=δ(∑v∈Nu⁡αuv(Whv))

where δ is a sigmod function.

This formula represents the propagation process through a graph. Each vertex calculates an attention weight with adjacent vertices, then smooths them and calculates a weighted sum of the adjacent vertex’s features to obtain the final output of that vertex. This procedure is repeated for each vertex in the graph.

GAT is a powerful graph model applicable to a wide range of graph-related problems. Its effectiveness in information extraction tasks is demonstrated by the following key strengths:

Cross-attention: GAT employs the attention mechanism to compute attention weights between neighboring vertices. This enables the model to concentrate on pivotal nodes, thereby generating higher-quality representations for each node. Additionally, the attention mechanism enables the model to adeptly manage large-scale graphs with substantial structural variations.

Learnable attention weights: Attention weights are trainable through a linear function, allowing the model to dynamically prioritize important vertices during training. This enhances GAT’s flexibility in determining the significance of vertices within the graph.

Ranking mechanisms: GAT frequently integrates a rank accumulation mechanism into its attention weight learning process. This augmentation strengthens the model’s proficiency in discerning the importance of connections between vertices within a graph, particularly within the realm of information extraction.

Efficiency and flexibility: GAT demonstrates the ability to manage intricate and evolving graphs without compromising on model efficiency. This attribute proves invaluable in tasks like information extraction, especially when dealing with intricate text graphs containing numerous relationships and information embedded within the vertices and edges.

Hence, we advocate for the adoption of GAT in our graph model, leading to notable outcomes. The ensuing section will delve into a detailed presentation of the results attained.

Loss function and optimization

We used cross-entropy loss (Mao, Mohri & Zhong, 2023) in GNNs classification task. This loss function is often favored in classification problems, especially when our model is faced with many different classes of objects.

Using the cross-entropy loss loss function helps us train our model so that it is capable of classifying graph objects accurately and efficiently. Specifically, cross-entropy loss is defined by the formula:

(10) LCE=−∑i=1N⁡yilog⁡pi

To optimize the loss function during model training, we use the Adam with Weight Decay (AdamW) (Loshchilov & Hutter, 2017) optimization algorithm. AdamW is a variant of Adam (Kingma & Ba, 2014), a gradient-based optimization method commonly used in machine learning and deep learning tasks.

AdamW was designed to solve Adam’s problem related to instability during training and unwanted growth of model weights. AdamW retains all the benefits of Adam, such as integrated adaptive learning rates and momentum, but adds a “weight decay” component.

Weight decay is a primary method to control overfitting in machine learning models by imposing a cost that depends on the weights of the parameters. In AdamW, the weight decay component is calculated and added to the weight update process. This helps prevent excessive growth of weights, minimizes the risk of overfitting, and improves the generalization ability of the model.

AdamW has demonstrated good performance in a variety of model training tasks and is generally popular among the research and development community in the field of machine learning. The combination of adaptive learning rates and weight decay helps improve model accuracy and stability, while minimizing the risk of overfitting during training.

In our study, we applied the AdamW optimization method to our GAT model. When working with large graphs like scanned image datasets, there is a high risk of overfitting due to the diversity of the data and the complexity of the graph. AdamW with weight decay component helps control excessive growth of weights, reduces the risk of overfitting and improves the generalization ability of the model. We describe our proposed ConBGAT model by pseudo-code in the Algorithm 1 below.

Algorithm 1 Pseudo-code of our proposed ConBGAT model.

Input: Scanned Image I	
Output: Extracted Information	
for k = 1 to I do	
   Text Feature Extraction using DistilBert in Eq. (1)	
 end for	
 Extract Text Feature TFE in Eq. (2)	
 for k = 1 to I do	
    Image Feature Extraction using ResNet50 in Eq. (3)	
 end for	
  Extract Image Feature IFE in Eq. (4)	
  Concat Text Feature and Image Feature ϒ¨ in Eq. (5)	
 for t = 1, 2, … , T do	
     Training model using GAT: ConBGAT = GAT( ϒ¨, p)	
     Calculate Loss Function in Eq. (10)	
     Optimize Loss Function with AdamW	
end for	
Return Extracted Information	

Implementation and experimental result

Dataset

We used the SROIE dataset (Huang et al., 2019) with 973 receipts from stores. In this dataset we used 626 invoices for training and 347 invoices for testing. Each invoice has four main text fields including: Company, Address, Date and Total. The dataset is mainly English characters and numbers, the dataset exhibits variable layouts and complex structures. To facilitate research, the dataset also includes annotations for each text bounding box, including their corresponding coordinates and records. Fig. 5A shows a sample of this dataset.

Figure 5 Some images from three datasets: (A) SROIE, (B) CORD and (C) FUNSD.

Consolidated Receipt Dataset for Post-OCR Parsing (CORD) (Park et al., 2019). CORD is a groundbreaking dataset designed for invoice analysis, marking a significant milestone in publicly available resources for this purpose. It comprises meticulously annotated invoices, catering to both optical character recognition (OCR) and parsing. This dataset encompasses 1,000 invoices, with 800 images allocated for the training set, 100 for validation, and another 100 for testing. The primary objective is to precisely categorize every word within the invoice into one of 30 fields across four distinct categories. Notably, our study utilized officially provided OCR images and annotations. The accompanying Fig. 5B offers a glimpse into the invoice samples within the dataset.

Form Understanding in Noisy Scanned Documents (FUNSD) (Jaume, Ekenel & Thiran, 2019) represents a publicly available dataset established to facilitate research and advancement in methods aimed at comprehending and extracting information from scanned forms plagued by noise. Encompassing a diverse array of form types such as applications, ballots, invoices, and more, this data stems from the RVL-CDIP dataset (Harley, Ufkes & Derpanis, 2015). Within the FUNSD dataset, there exist 199 real-world scanned forms, meticulously annotated to delineate 9,707 semantic entities. Among these, 149 images serve as training data, while 50 are earmarked for testing purposes. Offering versatility for a multitude of tasks, this dataset is particularly well-suited for the specific task undertaken in this study, which involves assigning each word a label selected from a predefined set of four categories: “Question,” “Answer,” “Header,” or “Other.” Table 1 shows the overview of three datasets: SROIE, CORD and FUNSD in our experiments and illustrated in Fig. 5C are some sample forms from these datasets.

Table 1 Overview of three datasets: SROIE, CORD and FUNSD in our experiments.

Dataset	Train	Test	
SROIE	626	347	
CORD	800	100	
FUNSD	149	50	

Data processing

First, the data is loaded and preprocessed, which involves reading bounding box coordinates from CSV files and loading the corresponding images. Next, a graph is constructed by grouping bounding boxes into lines based on vertical overlap, while horizontal and vertical connections are established between overlapping boxes.

Text and image features are then extracted. Text features include counts of uppercase letters, lowercase letters, whitespaces, alphabetic characters, numeric characters, and special characters within each bounding box. Image features capture the relative distances between connected bounding boxes, as well as features within each box, extracted using ResNet50.

The DistilBERT model is used to generate semantic embeddings for the text, capturing the contextual meaning of the text. Finally, features are integrated and labeled. The get_data function consolidates the extracted features and labels for all documents in the dataset. It iterates through each file, performing graph formation and feature extraction.

Numerical features (relative distances, text features, image features, etc.,) are concatenated with DistilBERT representations to create a comprehensive feature representation for each node in the graph. Numerical labels are assigned to each bounding box based on its predefined category (e.g., ‘company’, ‘address’, ‘date’, ‘total’, etc.,).

PyTorch Geometric Data objects are constructed, encapsulating the graph structure (nodes, edges), node features, and labels. Finally, the processed data is saved into training and testing datasets for subsequent model training.

This preprocessing pipeline transforms raw document images and bounding box information into graph-structured data with rich node features, ready for training on the ConBGAT information extraction task. It leverages both spatial relationships (graph structure) and semantic content (text representations) to enable the model to learn and make predictions about the roles of different text elements within documents.

Implementation details

We did experiments on Pytorch with GPU Nvidia GeForce GTX 1650Ti (4GB of memory). We train the GAT model with four layers. For the correction factor, we use Dropout with ratio 0.2 for GAT. For pre-train DistilBERT model, we used the default parameters on this model. The text embeddings has 512 dimension. Parameters of ResNet50 model are the same with (He et al., 2016). The fully connect layer is responsible for changing the output dimension 512. Our model is trained for about 90 min on 2,000 epochs, AdamW optimization function is used with learning rate 0.0001 for models to optimize cross-entropy loss and use batch size equal to 16 in the training phase.

Evaluation metrics

We used F1-score (Chase Lipton, Elkan & Narayanaswamy, 2014) to evaluate the performance and effectiveness of our model in a detailed and quantitative manner. For each scanned image in the test set, the extracted text is compared with reality. Extracted text is marked as correct if both the content and categories of the extracted text match reality; otherwise, it is marked as incorrect. F1-score is an index that evaluates the performance of a classification model. It is a composite index of precision and recall. Precision is defined as the ratio of true positive scores among all scores predicted by the model to be positive (TP + FP). Meanwhile, recall is defined as the ratio of True Positive scores among those that are actually positive (TP + FN).

(11) Precision=TPTP+FP

(12) Recall=TPTP+FN

(13) F1−score=2∗Precision∗RecallPrecision+Recall

where TP, FP, FN represent for True Positive, False Positive, False Negative.

Experimental results and discussion

Based on the method proposed above, we conduct experiments and evaluate our proposed ConBGAT model on three datasets. Our experiments are summarized as follows:

• Conducting experiments to select the best optimization algorithm in five algorithms for our proposed ConBGAT model.

• Comparing the selected GAT component in our proposed ConBGAT model with four GNN models on the SROIE dataset.

• Evaluating our proposed ConBGAT model against three previously introduced in and well-performing models on the same problem using the SROIE dataset.

• Assessing diversity across multiple datasets by comparing the performance of our proposed ConBGAT model with fifteen baseline models across three different datasets: SROIE, FUNSD, and CORD

Next session describes the results in detail.

Firstly, we conduct experiments to choose the best optimization algorithm for our proposed model. We use some optimization algorithms such as: SGD (Liu, Gao & Yin, 2020), RMSProp (Elshamy et al., 2023), Adagrad (Zhang, Lei & Zhao, 2018), Adam (Kingma & Ba, 2014), AdamW (Loshchilov & Hutter, 2017) to compare the result and find the best optimization algorithm. Table 2 shows the comparison results of these optimization algorithms based on two measures: F1-score and Loss. Based on the result shown in Table 2, we can see that AdamW achieved the highest F1-score (0.97), followed by Adam (0.94), RMSProp (0.90), SGD (0.86), and Adagrad (0.8643). Similarly, AdamW has the lowest loss (0.1488), followed by Adam (0.2876), RMSProp (0.3632), Adagrad (0.4868) and SGD (0.5688). The results show that AdamW outperforms the remaining methods in this case, with higher F1-score and significantly lower loss. This also proves that it can help improve model accuracy and generalization better than other optimization algorithms.

Table 2 Comparison results of five optimization algorithms based on F1-score and loss measurements.

Optimization functions	F1-score	Loss	
SGD	0.86	0.5688	
RMSProp	0.90	0.3632	
Adagrad	0.8643	0.4868	
Adam	0.94	0.2876	
AdamW	0.97	0.1488	

Next, we select some basic GNNs models to experiment and evaluate with the selected GAT component in our proposed ConBGAT model such as: GCN (Zhang et al., 2019), GraphSAGE (Hamilton, Ying & Leskovec, 2017), Graph Isomorphism Network (GIN) (Xu et al., 2018), simplifying graph convolutional networks (SGConv) (Wu et al., 2019). Table 3 shows the results of comparing the F1-score measure of the proposed ConBGAT model with other GNNs models on the SROIE dataset about Company entities, addresses, dates and total invoices. Specifically, our ConBGAT model has a F1-score measure 0.98, 0.98, 0.97 and 0.95 respectively for each entity: company, address, date and total invoice. The proposed ConBGAT model also achieved the highest accuracy for all entities, with Macro Average is 0.97. The result in Table 3 shows that the proposed model is an effective model for the task of entity classification on the real dataset. The model can learn complex relationships between entities, and has high accuracy for both each entity and all entities evaluation.

Table 3 F1-score results of four compared GNNs models with our ConBGAT model.

Entities	GCN	SAGE	SGConv	GIN	ConBGAT	
Company	0.8	0.85	0.68	0.73	0.98	
Address	0.8	0.88	0.73	0.76	0.98	
Date	0.76	0.8	0.77	0.68	0.97	
Total	0.72	0.75	0.8	0.71	0.95	
Macro average	0.77	0.82	0.745	0.72	0.97	

Next, to prove the effectiveness of the proposed model, we have selected three previously introduced and developed models that have achieved good results for this problem to compare and evaluate with our proposed method. Our compared models are spectral graph convolutional network (Hung, 2022), Bi-LSTM-CRF (Huang, Xu & Yu, 2015) and BERT-CRF (Souza, Nogueira & Lotufo, 2019).

For the result in Table 4, we used the result of the research (Hung, 2022) for spectral graph convolutional network and (Hua et al., 2020) for Bi-LSTM-CRF, BERT-CRF. In Table 4 shows that our ConBGAT model outperforms the baseline models on all components of the dataset SROIE. Specically, our proposed model ConBGAT has the highest F1-score for Company, Address and Date labels. For Total label, F1-score of our proposed model is higher than Bi-LSTM-CRF and BERT-CRF, but lower than Spectral GCN model only 0.01 score.

Table 4 F1-score results on each label of our proposed ConBGAT model with three compared baseline models on the SROIE dataset.

Entities	Spectral graph convolutional network	Bi-LSTM-CRF	BERT-CRF	ConBGAT	
Company	0.85	0.851	0.868	0.98	
Address	0.93	0.883	0.891	0.98	
Date	0.95	0.942	0.962	0.97	
Total	0.93	0.835	0.847	0.95	
Macro average	0.915	0.878	0.892	0.97	

Finally, to assess diversity across multiple datasets, we compared the performance of our proposed ConBGAT model with fifteen baseline models across three different datasets: SROIE, FUNSD, and CORD, Table 5 shows the results. In Table 5, we used the results of LayoutLMv3BASE, Spectral Graph Convolutional Network, BROS, PICK, DocTr, ESP and GenKIE models are supplied by Huang et al. (2022), Hong et al. (2020), Hung (2022), Liao et al. (2023), Yang et al. (2023) and Cao et al. (2023) all other models by Xu et al. (2020a, 2020b).

Table 5 F1-score results of our proposed ConBGAT model with the fifteen compared baseline models on the SROIE, FUNSD and CORD datasets.

Model	SROIE	FUNSD	CORD	
BERTBASE	90.99	60.26	89.68	
BERTLARGE	92.00	65.63	90.25	
UniLMv2BASE	94.59	68.90	90.92	
UniLMv2LARGE	94.88	72.57	92.05	
LayoutLMBASE	94.38	78.66	94.72	
LayoutLMLARGE	95.24	78.95	94.93	
LayoutLMv2BASE	96.25	82.76	94.95	
LayoutLMv2LARGE	96.61	84.20	96.01	
LayoutLMv2LARGE(excludingOCRmismatch)	97.81	–	–	
LayoutLMv3BASE	–	90.29	96.56	
SpectralGraphConvolutionalNetwork	91.50	-	–	
BROS	94.93	81.21	95.58	
PICK	96.79	–	–	
DocTr		84.00	98.20	
ESP	–	88.88	95.65	
GenKIE	97.40	83.45	95.75	
ConBGAT	97.40	89.61	96.72	

On the SROIE dataset, ConBGAT exhibits competitive performance compared to other models, achieving a high score of 97.40, only slightly behind LayoutLMv2LARGE(excluding OCR mismatch). This can be attributed to our ConBGAT model’s lower computational resource requirements and shorter training time compared to LayoutLMv2LARGE (excluding OCR mismatch). Despite these advantages, ConBGAT still demonstrates significantly competitive performance and is not significantly inferior to other LayoutLM models, highlighting ConBGAT’s strong capabilities in handling tasks related to information extraction and classification on invoices and documents.

On the FUNSD dataset, LayoutLMv3BASE exhibits the best performance with a score of 90.29. This highlights LayoutLMv3 ability to analyze layout and understand the spatial semantics of complex documents. ConBGAT, although not achieving the highest score, still shows competitive performance compared to other LayoutLM models. Additionally, ConBGAT is optimized in terms of computational resource efficiency and training time compared to LayoutLMv3.

On the CORD dataset, DocTr yields the best performance at 98.20. However, the parameters used for DocTr are around 153 M, while our ConBGAT only consumes 45 M of computational resources.

Overall, ConBGAT demonstrates superior or competitive performance across all three datasets, proving its effectiveness, high generalization capability, and efficient use of computational resources while still achieving competitive performance compared to other models. The combination of the GAT attention mechanism, ResNet50’s image feature extraction capability, and DistilBERT’s text processing ability has contributed to ConBGAT’s success.

Overall, ConBGAT demonstrates competitive performance across all three datasets, falling only slightly behind LayoutLM models with larger computational resources and longer training times. This proves the efficiency of the ConBGAT model, its high generalization ability, and its effective use of computational resources while still achieving competitive performance compared to other models. The combination of the GAT’s attention mechanism, ResNet50’s image feature extraction capability, and DistilBERT’s text processing ability contributed to ConBGAT’s success.

Detailed of results

AdamW proved to be the optimal optimizer, achieving the highest F1-score (0.97) and the lowest loss (0.1488). Compared to other algorithms such as SGD, RMSProp, Adagrad, and Adam, AdamW demonstrated superior capability in improving model accuracy and generalization.

The ConBGAT model was compared against other GNN models like GCN, GraphSAGE, GIN, and SGConv on the SROIE dataset for the following entities: company, address, date, and total amount. ConBGAT achieved the highest accuracy for all entities, with a Macro Average of 0.97. The results indicate that ConBGAT is an effective model for entity classification tasks on real-world datasets, capable of learning complex relationships between entities.

ConBGAT was also compared to Spectral Graph Convolutional Network, Bi-LSTM-CRF, and BERT-CRF models on the SROIE dataset.

On the three datasets SROIE, FUNSD, and CORD, ConBGAT consistently produced competitive F1-scores compared to other models, while also being optimized in terms of computational parameters and training time. The combination of the GAT’s attention mechanism, ResNet50 for image feature extraction, and DistilBERT for text embedding yielded positive results for the model.

When looking on the results shown in Tables 3–5, it can be seen that our proposed method produces better results than other models. Thanks to the attention mechanism of GAT combined with Resnet50 to extract features for images and DistilBERT used to perform text embeddings, since then, the model has achieved positive results. Below are some images in the test dataset where we extracted entities shown in Fig. 6.

Figure 6 Some successful extracted entity results from scanned image in our experiments.

However, when analyzing in detail, we can see that all labels in our method produce superior results compared to the rest, but only in the Total label, our model has lower results. compared to the model spectral graph convolutional network (He et al., 2023) to explain this problem, there are a few points as follows: First is the difference in labels, there is a quite large difference between labels in the data. Second, the Total label is quite limited in being able to extract detailed features, because it only contains very few numbers, making feature extraction more difficult and the above result tables also reflect clearly about this. The Total label always produces lower results than all other labels.

While our results and proposed method offer significant potential in addressing practical challenges associated with textual information identification, they also come with certain limitations:

Generalization: The model may struggle to generalize to new datasets. This limitation occurs because the model is trained on a specific dataset; if the characteristics of a new dataset differ significantly from the training data, the model may yield inaccurate results.

Data characterization challenges: The process of characterizing data can face several limitations, such as unbalanced labels and information containers. This can cause the model to learn inaccurate features or fail to fully capture the true characteristics of the data. Additionally, some containers may hold insufficient information about their attributes, resulting in inaccurate predictions for those containers.

These limitations present opportunities for future research and development aimed at enhancing and expanding the application of deep learning methods in the field of information identification from scanned images. In Fig. 7 we show some incorrectly predicted samples.

Figure 7 Some failed extracted entity results from scanned image in our experiments.

The “Total” field in invoices presents a significant challenge for accurate extraction due to its inherent variability. This field often appears in different positions within the document, utilizes diverse formats for representing monetary values, and shares semantic similarities with other fields containing the word “Total.” These factors contribute to frequent misclassifications, hindering the overall performance of information extraction models.

To mitigate these challenges, a multi-faceted approach is proposed. First, data augmentation techniques can be employed to expand the training dataset with a wider range of “Total” field variations, enabling the model to learn and generalize better. Second, introducing data variations such as rotation, brightness adjustments, and resizing can further enhance the model’s robustness to different invoice layouts and styles. Lastly, post-processing rules or language models can be applied to filter and standardize the predicted results, ensuring consistency and accuracy in the extracted “Total” values.

Conclusion

In conclusion, our research has successfully introduced the ConBGAT model, a novel approach that leverages the power of CNN, DistillBert, and GAT architectures to effectively address the complex task of information extraction from scanned images. By integrating these advanced techniques, we have demonstrated a significant improvement in accuracy and robustness compared to existing methods, as evidenced by our extensive experiments on three diverse datasets.

However, we acknowledge that the “Total” field remains a particularly challenging aspect of information extraction due to its inherent variability in format and position within invoices. To overcome this limitation, we propose further research into data augmentation techniques, incorporating a wider range of “Total” field variations to enhance the model’s learning capacity. Additionally, exploring data variation methods like rotation and resizing could improve the model’s adaptability to diverse invoice layouts.

Furthermore, we recognize the potential of post-processing techniques, such as rule-based filtering or language models, to refine and standardize the extracted results, ensuring greater accuracy and consistency. By integrating these enhancements, we anticipate a substantial improvement in the model’s ability to accurately identify and extract the “Total” field, even in complex and varied invoice documents.

Moving forward, we are committed to refining our model by optimizing its performance and reducing computational complexity. We also aim to explore the integration of multitasking capabilities, enabling the model to simultaneously address multiple information extraction tasks within a single image. By continuously pushing the boundaries of this research, we strive to develop a comprehensive and practical solution for automated information extraction from scanned imagedocuments, with a particular focus on overcoming the challenges posed by the “Total” field.

Supplemental Information

Supplemental Information 1 CORD Dataset.

Additional Information and Declarations

Competing Interests

Author Contributions

Data Availability

The authors declare that they have no competing interests.

Duy Ho Vo Hoang performed the experiments, performed the computation work, prepared figures and/or tables, and approved the final draft.

Huy Vo Quoc analyzed the data, prepared figures and/or tables, and approved the final draft.

Bui Thanh Hung conceived and designed the experiments, performed the experiments, performed the computation work, prepared figures and/or tables, authored or reviewed drafts of the article, and approved the final draft.

The following information was supplied regarding data availability:

The CORD raw data are available in the Supplemental File. They came from CORD Dataset: Park, S., Shin, S., Lee, B., Lee, J., Surh, J., Seo, M., & Lee, H. 2019. CORD: a consolidated receipt dataset for post-OCR parsing. In Workshop on Document Intelligence at NeurIPS 2019.

The FUNSD Dataset is available at: Jaume, G., Ekenel, H. K., & Thiran, J. P. (2019, September). Funsd: A dataset for form understanding in noisy scanned documents. In 2019 International Conference on Document Analysis and Recognition Workshops (ICDARW) (Vol. 2, pp. 1-6). IEEE. https://doi.org/10.48550/arXiv.1905.13538.

The SROIE Dataset is available at: Huang, Z., Chen, K., He, J., Bai, X., Karatzas, D., Lu, S., & Jawahar, C. V. (2019, September). Icdar2019 competition on scanned receipt ocr and information extraction. In 2019 International Conference on Document Analysis and Recognition (ICDAR) (pp. 1516-1520). IEEE. https://doi.org/10.1109/ICDAR.2019.00244.

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
