# Peer review of "ConBGAT: a novel model combining convolutional neural networks, transformer and graph attention network for information extraction from scanned image"

_PeerJ Computer Science, doi:10.7717/peerj-cs.2536_

## Round 0.1 · original submission · Major Revisions

Dear authors,

Thank you for submitting your manuscript. Reviewers have now commented on your article and suggest major revisions. Reviewer 1, Reviewer 4, and Reviewer 5 have asked you to provide specific references. You are welcome to add them if you think they are relevant. However, you are under no obligation to include them, and if you do not, it will not affect my decision. When submitting the revised version of your manuscript, it will be better to address the following:

1- The research gaps and contributions should be clearly summarized in the introduction section. Please evaluate how your study is different from others. Contributions listed in the Introduction section do not seem real “contributions”. Some of them mentions the performed works for experimentation. Please correct this list.
2- The values for the parameters of the algorithms selected for comparison should be given.
3- The paper lacks the running environment, including software and hardware. The analysis and configurations of experiments should be presented in detail for reproducibility. It is convenient for other researchers to redo your experiments and this makes your work easy acceptance. A table with parameter settings for experimental results and analysis should be included in order to clearly describe them.
4- Advantages and disadvantages of the methods should be clarified. What are the limitations of the method(s) used in this paper? Please indicate the practical advantages and discuss the limitations of the research.
5- Equations should be used with correct equation number. Please do not use “as follows”, “given as”, etc. Explanation of the equations should also be checked. All variables should be written in italic as in the equations. Their definitions and boundaries should be defined. Necessary references should be provided.

Best wishes,

Reviewer 1 ·

Basic reporting

The authors proposed a model combining CNN, BERT, and graph attention network for information extraction from scanned document images. Experiments were done on the SROIE, FUNSD, and CORD datasets.

Experimental design

A model named ConBGAT is proposed for extraction of information from scanned document Images. The ConBGAT model combines CNN in and Transformer-DistilBERT which extracts features. Employing the Graph Attention Network (GAT) model to glean insights from the graph structure, enabling the model to comprehend the intricate relationships among components.

Validity of the findings

The following are some concerns that should be addressed:

1) A brief literature review in this regard is required starting from the classical approaches to machine learning-based approaches.

2) What is the motivation of combining CNN in and Transformer-DistilBERT? A mathematical explanation is needed in this regard. It can be added at the end of Literature review or in the proposed approach section.

3) The experiments are not adequate. The proposed method lacks comparison with other well-known methods and cannot show its superiority. The comparison shown only on different classifiers. The results on the datasets are not convincing. Details about the dataset are missing, such as the procedure of collection of dataset, and labeling/annotation of dataset.

4) The future direction and limitations of the proposed work need to be discussed in the Conclusion.

Reviewer 2 ·

Basic reporting

1. In the abstract, abbreviations that appear for the first time in the article need to be used in full, so please check them.
2. Most of the references cited in the article are formatted incorrectly.
3. In the Introduction section, the advantages and disadvantages of deep learning as well as traditional methods are briefly described, and no mention is made of the current problems encountered in feature extraction for scanned images.
4. In response to the innovation of this paper, the article lists a number of articles, but in the final analysis it only proposes a model.
5. In lines 153-160, in this paragraph it is mentioned that CNN and DistillBERT models are used for image and text features and trained on CNN. Wasn't the GAT model used throughout? The article doesn't state exactly how many modules are included in the proposed model and what each module does.
6. In Word Recognition and Feature Extraction, more and more details of the models proposed in this paper should be presented rather than analyzing all the methods, which should be presented in related work.
7. Not all of these datasets listed in the datasets section are used in your experiments; it is sufficient to present only the datasets used in this paper.
8. The experimental part is too easy.
9. Detailed captions should be added to all images in the paper.

Experimental design

As above

Validity of the findings

As above

Additional comments

As above

·

Basic reporting

The paper combined three modules CNN, BERT and graph attention network for scanned image recognition. The authors applied their proposed model on three benchmark datasets and achieved good performance using the F1-score evaluation metric. However, my comments for improving the quality of the paper are as follows:

1- First of all, the English used in the entire paper should be improved. The sentences and subsections are not coherent.

2- Authors should provide the following sections in the paper: a related work and background theory.
2.1- The authors must provide details of the different modules (CNN, Transformer, BERT, etc) in the background theory section.
2.2- Since the proposed architecture is related to the scene text detection and recognition (STDR) problem, authors should provide a literature review of the recent and state-of-the-art (SOTA) models of  STDR.
2.3- What is the rationale behind using different module networks in the paper? Why did the authors integrate these three specific modules? Why the authors only have not used a complete CNN or Transformer network as in to address the problem?
2.4- The abstract needs a major revision and has the following shortcomings:
2.5- The introduction is not well written. The first paragraph is too long.  It should include. Many deep-learning models can be used in lines 37-38. Please cite more related references here.
2.6- Some of the contributions stated in lines 61-76 are not contributions of this paper. For example, CNN and Transformer-DistillBERT are the only ones utilized in this paper.
2.7- Why did the authors not use a complete transformer decoder as in [4] as decoder after extraction of the features in the paper? It would be great if the authors provided an ablation study of it.
2.8- All the optimization algorithms used in the paper have also been already propped and cannot be counted as a contribution in line 72. There are plenty of object detection and scent text detection and recognition methods that use ADAMW as a very good optimizer.
2.9- Citations are missing for SROIE, FUNSD, and CORD in lines 74-75.
2.10- Authors should provide more details in the caption of the figures and tables. The provided captions are short for all the figures and tables.

3-  lines 84-86: Rewrite or move this paragraph into the introduction. The last sentence needs a citation. 
3.1- Please create a related work section and move the lines 88-194 into the related word section.

Experimental design

What are the modifications to the utilized modules in Figure 1? What are the main contributions to the proposed architecture? It would be good if the authors highlighted them.

The presented architectures in Figure 1 and Figure 2 lack of novelty. The descriptions of the leveraged sub-blocks in the “methods and materials” section are also incomplete and incoherent.
 
What is the word embedding techniques in line 205? Please provide more details of it.
It would be also good if the authors explained how the input image passes through the network, and what happens to the feature dimensions in every stage of modules.
It’s hard to understand the lines 216-250. Please rewrite them.
Correct the “As introduced above” in line 253 by inserting a comma.
The structure of subsections in the  “methods and materials” section has not been organized well. One example, bolding the “Cross-Attention” in line 309.
• The Algorithm 1 is not new and it already has been introduced. It would be good to cite it.

Validity of the findings

It is recommended to move the “dataset (line 361)” into the results section.
Rename the “hyperparameters (line 389)” into the implementation details and move it also to the results section.
What is the purpose of the authors for using  “to use fix parameters “in line 392? What are the fixed parameters?
In the section of line 389, how long does it take the model to be trained? How many images have been used? Have the authors used pre-training models? Have the authors used any synthetic datasets for training? 
The results have not been well discussed. Please provide a discussion and ablation study subsections in the result section and provide more details about the results. The authors only reported the results without any justification.
In the “limitation” subsection, authors should provide the following: some example images that show the failure cases. Why did that failure happen? And how or what modules can be integrated into the current model to address this failure?
- Many STDR methods achieved good performance in benchmarks, and they are publicly available to test. It would be good if the authors also compared their model with those methods.

Reviewer 4 ·

Basic reporting

1. The entire work is written in professional, clear English that is simple to understand and use short sentences.
2. A comprehensive background is given in the introduction, which also highlights the drawbacks of earlier techniques and the significance of information extraction from scanned pictures. The background review is thorough and includes plenty of references.

Experimental design

1. The research question is well-defined, addressing the need for improved accuracy and efficiency in information extraction. The paper clearly states how it fills the identified knowledge gap.

2. The experimental design is robust, with detailed descriptions of the model architecture, datasets used, and evaluation metrics. The methods are described in sufficient detail to enable replication.

3. here is no indication of ethical issues. The datasets used (SROIE, FUNSD, and CORD) are standard benchmarks in the field.

Validity of the findings

1. The underlying data provided are robust and statistically sound. The datasets used are well-known and provide a solid basis for comparison.

2. The statistical methods used to evaluate the model's performance are appropriate and correctly applied. The results are presented with appropriate metrics (e.g., accuracy, precision, recall, F1 score).

3. The conclusions are well-supported by the results. The paper successfully demonstrates that the ConBGAT model outperforms existing methods on multiple datasets, highlighting its effectiveness.

Additional comments

1. The major strength of the paper is the innovative combination of CNN, BERT, and GAT models, which effectively leverages both image and text features. The comprehensive experimental evaluation across multiple datasets further strengthens the findings.

2. Provide more details on the preprocessing steps and parameter tuning.

3. Discuss potential limitations of the proposed approach and possible future work to address these limitations.

·

Basic reporting

The authors describe a model called ConBGAT for information extraction in scanned documents, which integrates convolutional neural networks (CNNs), transformers, and attention graph networks (GNN). GNNs are constructed from regions of the text obtained by OCR and by the use of CNN and DistilBERT for information extraction. In GNN, two features for the nodes are used: image features and lexical features. Two deep learning models are employed to extract features: CNN model for extracting features from images and DistilBERT model for extracting text content features in the bounding box.
Detailed remarks
• Line 222, “Where” should not be justified and can start by a lowercase letter, similarly in line 264, 291, 296, 301

Questions
• What is the definition of a sentence in this context?
• Line 246-248: Did you use other combination operation apart the “+”?
• Line 297, this sentence doesn’t have a verb
• Many bibliographic references of Djedjiga Belhadj are missing showing similar techniques and better results on SROIE:
o Belhadj, D., Belaïd, Y., & Belaïd, A. (2021, September). Automatic generation of semistructured documents. In Document Analysis and Recognition–ICDAR 2021 Workshops, Proceedings, Part II 16 (pp. 191-205). Springer International Publishing.
o Belhadj, D., Belaïd, Y., & Belaïd, A. (2021, September). Consideration of the Word’s Neighborhood in GATs for Information Extraction in Semi-structured Documents. In Document Analysis and Recognition–ICDAR 2021, Proceedings, Part II 16 (pp. 854-869). Springer International Publishing.
o Belhadj, D., Belaïd, A., & Belaïd, Y. (2023, August). Improving Information Extraction from Semi-structured Documents Using Attention Based Semi-variational Graph Auto-Encoder. In International Conference on Document Analysis and Recognition–ICDAR 2023 (pp. 113-129). Cham : Springer Nature Switzerland.
o Belhadj, D., Belaïd, A., & Belaïd, Y. (2023, September). Low-Dimensionality Information Extraction Model for Semi-structured Documents. In International Conference on Computer Analysis of Images and Patterns–CAIP 2023 (pp. 76-85). Cham : Springer Nature Switzerland.
o Belhadj, D., Belaïd, A., & Belaïd, Y. (2024, February). Homomorphic encryption friendly Multi-GAT for information extraction in business documents. In The International Conference on Pattern Recognition Applications and Methods–ICPRAM 2024.

• It is not clear how many fields are extracted in CORD and FUNSD. Are the percentages given in tables 2, 3 and 4 an average for the different fields or represent the best
• This model apparently only works under supervision. How to deal with unknown documents?
• No word is said about the tagging of the CORD and FUNSD datasets. Can you tell how this was done?

Experimental design

No comment

Validity of the findings

No comment

Additional comments

Several information extraction systems have been published with comparison of the results on SROIE. There is even an international competition which took place at ICDAR, of which we cannot find any trace here. It is recommended to study these works and compare them against your works

---

## Round 0.2 · Minor Revisions

Dear authors,

Thank you for your paper. Some of the prior reviewers did not respond to the invitation to re-review. According to one reviewer, your paper still needs revision and we encourage you to address the concerns and criticisms of Reviewer 5 and resubmit your article once you have updated it accordingly.

Best wishes,

·

Basic reporting

The authors addressed my previous comments.

Experimental design

The authors addressed my previous comments.

Validity of the findings

The authors addressed my previous comments.

·

Basic reporting

No comment

Experimental design

As I said in my previous review, the article lacks depth and technicality in the description of the method investigated. Moreover, similar works have been published on the same subject and the authors have not taken them into account.

Validity of the findings

It is still not clear what this work brings compared to the existing ones. The comparisons are limited to a few works and do not cover everything. I suggested several articles using the same techniques and obtaining good results, but the authors only kept from the list the article that touches on the good technique and omitted the articles that talk about the real method.

---

## Round 0.3 · Major Revisions

Dear authors,

Thank you for the revision. According to Reviewer 5, your paper still needs a revision on the performance comparison and we encourage you to address the concerns and criticisms of this reviewer and resubmit your article once you have updated it accordingly.

Furthermore, Reviewer 1 has asked you to provide specific references. You are welcome to add them if you think they are useful and relevant. However, you are not obliged to include these citations, and if you do not, it will not affect my decision.

Best wishes,

Reviewer 1 ·

Basic reporting

A brief literature review in this regard is required starting from the classical approaches to machine learning-based approaches. A few are mentioned here:

a) A. Dutta, A. Garai and S. Biswas, "Segmentation of Meaningful Text-Regions from Camera Captured Document Images," 2018 Fifth International Conference on Emerging Applications of Information Technology (EAIT), Kolkata, India, 2018, pp. 1-4, doi: 10.1109/EAIT.2018.8470403

b) Gatos, B., Louloudis, G., Stamatopoulos, N.: Segmentation of historical handwritten documents into text zones and text lines. In: 2014 14th International Conference on Frontiers in Handwriting Recognition (ICFHR), pp. 464–469. IEEE (2014)

c) Kaddas, P., Gatos, B., Palaiologos, K., Christopoulou, K., Kritsis, K. (2023). Text Line Detection and Recognition of Greek Polytonic Documents. In: Coustaty, M., Fornés, A. (eds) Document Analysis and Recognition – ICDAR 2023 Workshops. ICDAR 2023. Lecture Notes in Computer Science, vol 14194. Springer, Cham. https://doi.org/10.1007/978-3-031-41501-2_15

Experimental design

The authors addressed my previous comments.

Validity of the findings

The authors addressed my previous comments.

Additional comments

no comment

·

Basic reporting

Nothing to report

Experimental design

Nothing to report

Validity of the findings

Nothing to report

Additional comments

The results shown in Table 5 are incomplete or do not match reality. One example is LayoutLMv2LARGE which gives for SROIE: 0.9781 and not 96.61, which exceeds the performance of the authors' system which is: 97.40. Other systems in the literature perform better on SROIE. It would be important to mention these systems. Here is a table of what I know:

System Params F1 score
LAMBERT[9] 125M 98.17
LayoutLMV2(L)[27] 426M 97.81
LayoutLMV2(B)[27] 200M 96.25
StrucText[17] 107M 96.88
ViBERTGrid[18] 157M 96.40
TRIE[31] – 96.18
PICK[30] – 96.12
LayoutLM(L)[28] 343M 95.24
LayoutLM(B)[28] 113M 94.38
BERT(B)[7] 340M 92
Ours 41M 97.94 (Belhadj et al. ICDAR 2023)

[17] Li, Y., Qian, Y., Yu, Y., Qin, X., Zhang, C., Liu, Y., Yao, K., Han, J., Liu, J., Ding,E.: Structext: Structured text understanding with multi-modal transformers. In:Proceedings of the 29th ACM International Conference on Multimedia. pp. 1912–1920 (2021)

[18] Lin, W., Gao, Q., Sun, L., Zhong, Z., Hu, K., Ren, Q., Huo, Q.: Vibertgrid: a
jointly trained multi-modal 2d document representation for key information extraction from documents. In: International Conference on Document Analysis andRecognition. pp. 548–563. Springer (2021)Improving information extraction in SSDs using VGAE 17

[27] Xu, Y., Xu, Y., Lv, T., Cui, L., Wei, F.,Wang, G., Lu, Y., Florencio, D., Zhang, C.,Che, W., et al.: Layoutlmv2: Multi-modal pre-training for visually-rich documentunderstanding. arXiv preprint arXiv:2012.14740 (2020)

---

## Round 0.4 · accepted · Accept

Dear Authors,

Thank you for the revision. The reviewers of the preceding round did not respond to the invitation to review the current version. However, the paper has been sufficiently improved, and it now seems acceptable for publication. It is also important to note that particular attention should be paid to the correct formation of sentences in equations prior to the production stage. Many of the equations are part of the related sentences. Equations should be used with correct equation number. Please do not use “as follows”, “given as”, etc. Explanations of the equations should also be checked. All variables should be written in italic as in the equations. Their definitions and boundaries should be defined. "Extractioon" must be corrected before production steps.

Best wishes,